

# Prevalence of chronic obstructive pulmonary disease at high altitude: a systematic review and meta-analysis

Huaiyu Xiong[1,2,3], Qiangru Huang[1,2,3], Chengying He[2], Tiankui Shuai[1,2,3], Peijing Yan[3,4], Lei Zhu[1,2], Kehu Yang[3,4,5,6] and Jian Liu[1,2]

[1] The First Clinical Medical College of the First Hospital of Lanzhou University, Lanzhou, China
[2] Department of Intensive Care Unit, The First Hospital of Lanzhou University, Lanzhou, China
[3] Evidence-Based Medicine Center, School of Basic Medical Sciences, Lanzhou University, Lanzhou, China
[4] Institute of Clinical Research and Evidence Based Medicine, The Gansu Provincial Hospital, Lanzhou, China
[5] Evidence Based Social Science Research Center, Lanzhou University, Lanzhou, China
[6] Key Laboratory of Evidence Based Medicine and Knowledge Translation of Gansu Province, Lanzhou, China

Corresponding author
Jian Liu, medecinliu@sina.com

## ABSTRACT

**Background and objective:** Recently, several studies have investigated the prevalence of chronic obstructive pulmonary disease (COPD) at high altitude (>1,500 m). However, much remains to be understood about the correlation between altitude and COPD. We aimed to summarize the prevalence of COPD at high-altitudes and find out if altitude could be a risk factor for COPD.

**Methods:** We searched PubMed/Medline, Cochrane Library, Web of Science, SCOPUS, OVID, Chinese Biomedical Literature Database (CBM) and Embase databases from inception to April 30th, 2019, with no language restriction. We used STATA 14.0 to analyze the extracted data. A random-effect model was used to calculate the combined OR and 95% CI. Heterogeneity was assessed by the $I^2$ statistic versus $P$-value. We performed a subgroup analysis to analyze possible sources of heterogeneity. The Egger's test and the Begg's test were used to assess any publication bias.

**Results:** We retrieved 4,574 studies from seven databases and finally included 10 studies (54,578 participants). Males ranged from 18.8% to 49.3% and the population who smoked ranged from 3.3% to 53.3%. The overall prevalence of COPD at high-altitude was 10.0% (95% CI [0.08–0.12], $P$ < 0.001). In a subgroup analysis, based on different regions, the results showed that the prevalence in Asia was higher than that in Europe and America. Seven studies compared the relationship between the prevalence of COPD at high-altitudes and the lowlands. The results showed that altitude was not an independent risk factor for the prevalence of COPD ($OR_{adj}$ = 1.18, 95% CI [0.85–1.62], $P$ = 0.321). There was no publication bias among the studies.

**Conclusions:** Our study found a higher prevalence of COPD at high-altitudes than those from average data. However, altitude was not found to be an independent risk factor for developing COPD (PROSPERO Identifier: CRD42019135012).

## INTRODUCTION

Chronic obstructive pulmonary disease (COPD) is a prevalent and non-curable disease that causes heavy social and economic burden (*GBD 2015 Chronic Respiratory Disease Collaborators, 2017*). Evidence shows that COPD affects an increasingly younger age group (*Blanco et al., 2006*), accounting for 3.2% of males and 2.0% of females (*Brakema et al., 2019*). Additionally, it is the third leading cause of mortality in the world, accounting for 5.7% of all-death population (*Brakema et al., 2019*). In spite of this, COPD is still undertreated and underdiagnosed (*Burtscher, 2014*).

Previous studies have reported that globally, there are more than 400 million people living at high-altitudes (*Caballero et al., 2008*), however, the relationship between the prevalence of COPD and altitude is still unknown. Several studies have also demonstrated a negative correlation between COPD and altitude (*Coultas, Samet & Wiggins, 1984*; *Menezes et al., 2005*; *Laniado-Laborin et al., 2012*; *Horner et al., 2017*). For example, *Horner et al. (2017)* found a prevalence of 8.5% at high-altitudes and 9.9% in the lowlands for COPD. *Caballero et al. (2008)* reported that, with the increase in altitude, the prevalence of COPD increased. Other studies have reported that people living at higher altitudes may have a higher prevalence of COPD (*Hwang et al., 2018*; *Gaviola et al., 2016*; *Ezzati et al., 2012*; *Coté et al., 1993*). A study from Kyrgyzstan found that the prevalence of COPD at high-altitude was three-fold more than that in the lowlands (*Brakema et al., 2019*). Another review also suggested that people living in high-altitude areas may be at lower risk of developing COPD after immigrating to lower altitudes (*Burtscher, 2014*). Therefore, it is essential to clarify whether people living in high altitudes have a higher risk of COPD.

To the best of our knowledge, there has been no systematic review or meta-analysis on COPD at high-altitude and its prevalence. In addition, high-quality meta-analysis has been increasingly regarded as one of the key tools for establishing evidence (*Tian et al., 2017*). Therefore, we have analyzed previous studies to conduct a meta-analysis and summarize the prevalence of COPD at high-altitudes and explore whether altitude is an independent risk factor affecting the prevalence of COPD.

## MATERIALS AND METHODS

All the methods followed the PRISMA guidelines for conducting systematic review and meta-analysis. This meta-analysis was registered on the PROSPERO International Prospective Register of systematic reviews (PROSPERO Identifier: CRD42019135012).

### Data sources and searches

We searched PubMed/Medline, Cochrane Library, Web of Science, SCOPUS, OVID, Chinese Biomedical Literature Database (CBM) and Embase databases from inception to April 30th, 2019 to find out original studies that described the prevalence of COPD in high-altitude residents. There was no language restriction during our search process, and

we searched references of all relevant studies to make our search more comprehensive. The keywords and MESH terms were combined, and the search strategy we used in PubMed is as follows: ("Altitude" [Mesh] OR altitudes OR plateau) AND ("Pulmonary Disease, Chronic Obstructive" [Mesh] OR COPD OR Chronic Obstructive Pulmonary Disease) OR ("Lung Diseases, Obstructive" [Mesh] OR Lung Disease, Obstructive OR Obstructive Lung Disease) OR (Bronchitis, Chronic OR Chronic Bronchitis) OR ("Pulmonary Emphysema" [Mesh] OR Emphysemas, Pulmonary OR Pulmonary Emphysemas). The detailed search strategies from databases are shown in Appendix S1.

Studies that met the following inclusion criteria were included:

1. Studies that described the prevalence of COPD at high altitudes.
2. Studies that provided the Odds Ratio (OR) and 95% confidence interval (CI) or had enough data to calculate these.
3. The included subjects who had not been previously diagnosed as COPD.
4. Studies that included two or more participants.

The exclusion criteria included:

1. Studies that described other types of diseases with high-altitude, such as cardiovascular disease, asthma, acute respiratory distress syndrome (ARDS), and sleep apnea.
2. Studies that included animal researches, reviews, case reports, letters, and commentaries
3. Subjects diagnosed as COPD before being included in the research.

## Data extraction and quality assessment

Two authors (X.H.Y and H.Q.R) screened the titles and/or abstracts from retrieved studies independently, read the full text of eligible studies and extracted information from the original studies. The relevant information included: first author, published year, region, type of study, elevation from sea-level that the eligible participants lived at, body mass index (BMI), diagnostic criteria for COPD, sample size, mean age, gender, percentage of current smoking status, history of tuberculosis (TB), education level, corresponding to the adjusted OR value of the 95% CI, Variables of adjusted OR and prevalence of COPD at high-altitude and plains. Any difference in opinion was discussed between the two authors.

Variables of adjusted OR include household air pollution (HAP), age, gender, educational level, pack years, exposure to gas or dust, history of TB, BMI and ethnic origin. We defined high altitude as >1,500 m above sea level (*Cohen & Small, 1998*). The diagnosis for COPD in the studies met the following four criteria: post-bronchodilator forced expiratory volume in 1 s ($FEV_1$)/forced vital capacity (FVC) ratio less than 0.70, $FEV_1$/FVC less than 5% of age-dependent lower limit of normal (LLN), the International Classification of Disease definition (ICD), and patient-reported COPD (*Caballero et al., 2008*; *Halbert et al., 2006*; *Havryk, Gilbert & Burgess, 2002*; *Culver, 2012*; *Wurst et al., 2017*).

Quality assessment was done by the two authors individually (X.H.Y and H.Q.R). The Newcastle-Ottawa Scale (NOS) (*Peterson et al., 2011*), which contained 3 main

concepts: selection, comparability, and outcome assessment which were used to assess the quality of the cohort studies, and the Agency for Healthcare Research and Quality (AHRQ), which included 11 terms, was used to assess the quality of the cross-sectional studies.

## Data analysis

We used STATA 14.0 to analyze the extracted data. We calculated the unadjusted prevalence of COPD by dividing the number of cases by the total number of participants. We used the Review Manager (Version 5.3; Revman, Copenhagen, Denmark) to calculate the OR and the 95% CI when they were not available in the studies (*DerSimonian & Laird, 2015*). A random-effects model was used to calculate the combined adjusted OR and 95% CI (*DerSimonian & Laird, 2015*). Heterogeneity was assessed by the $I^2$ statistic versus $P$ value (*Higgins & Thompson, 2002*). A $P$-value of $\leq 0.05$ and $I^2 \geq 50\%$ were considered high heterogeneity and $I^2 \leq 50\%$ indicated heterogeneity in an acceptable range. We performed a subgroup analysis to analyze any possible source of heterogeneity when the heterogeneity was high. A sensitivity analysis was performed to detect if the results were stable and reliable. The Egger's test and the Begg's test were used to assess publication bias (*Song & Gilbody, 1998*). A funnel plot depicted when the studies were more than 10 (*Lau et al., 2006*). We considered a $P < 0.05$ as statistically significant.

## RESULTS

We finally retrieved 4,574 studies from 7 databases and additional records identified through other sources. After removing duplicates, 3,145 studies remained. We screened the titles and abstracts of the studies and 174 studies remained. We viewed the full texts of the 174 studies, and finally included 10 studies for this analysis. The specific steps are shown in Fig. 1.

A total of 54,578 participants were included from 10 studies and the sample size for a single study varied from 365 to 30,874. The characteristics of the included studies are shown in Table 1. Out of the 10 studies, 7 were from the Americas (Colombia, Mexico and Peru) (*Caballero et al., 2008*; *Menezes et al., 2005*; *Laniado-Laborin et al., 2012*; *Miele et al., 2018*; *Jaganath et al., 2015*; *Zaeh et al., 2016*; *Urrunaga-Pastor et al., 2018*), 1 was from Europe (Austria) (*Horner et al., 2017*), and 2 were from Asia (Kyrgyzstan) (*Brakema et al., 2019*; *Vinnikov, Brimkulov & Redding-Jones, 2011*). Eight studies met the fixed-ratio diagnostic criteria (*Brakema et al., 2019*; *Caballero et al., 2008*; *Menezes et al., 2005*; *Laniado-Laborin et al., 2012*; *Miele et al., 2018*; *Jaganath et al., 2015*; *Zaeh et al., 2016*; *Vinnikov, Brimkulov & Redding-Jones, 2011*), 1 met the definition of LLN (*Horner et al., 2017*) and 1 met the patient-reported criteria (*Urrunaga-Pastor et al., 2018*).

Among the participants, the proportion of males ranged from 18.8% to 49.3%, the proportion who smoked ranged from 3.3% to 53.3%. The subjects with a history of TB varied from 0.8% to 3.1%. The proportion with a high school level of education ranged from 0.6% to 21.8%. All 10 were cross-sectional studies. The results of the quality analysis were as follows: AHRQ scores ranged from 5 to 10 out of 10 cross-sectional

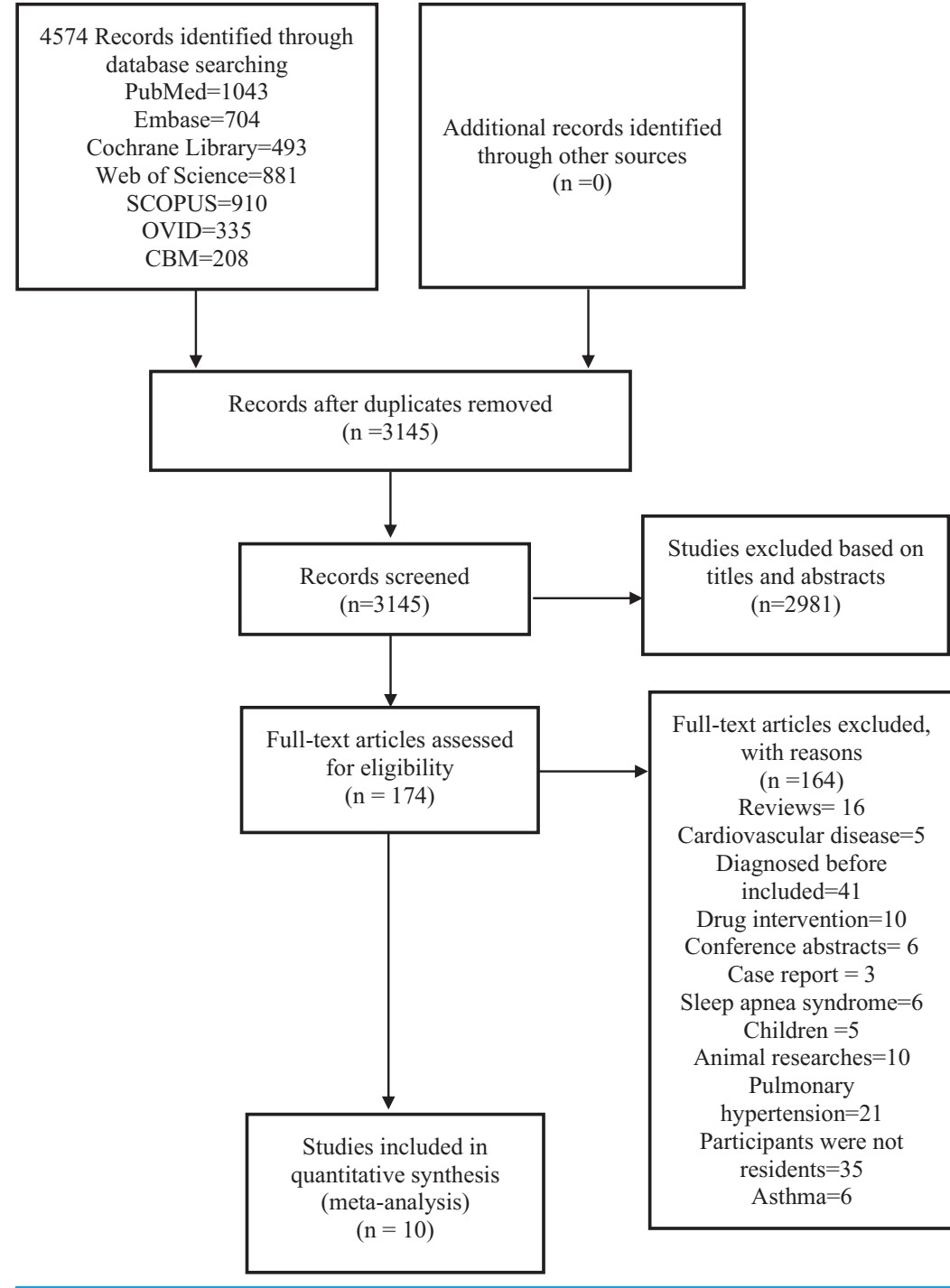

**Figure 1 PRISMA (preferred reporting items for systematic reviews and meta-analyses) flow diagram and exclusion criteria.**

studies. Six studies had a low risk of bias, and 4 had a moderate risk of bias; no study had a high risk of bias.

All of the 10 included studies described the prevalence of COPD at high-altitude. The results showed that the overall prevalence of COPD at high-altitude was 10.0%

**Table 1 Characteristics of included studies (n = 10).**

| Study | Year | Country (period) | Type | Height highland/lowland (m) | BMI (kg/m²) | Diagnostic criteria | N | Male (%) | Age (years) | History of TB (%) | Current smoking (%) | Higher education (%) | AHRQ | Variables of adjusted OR |
|---|---|---|---|---|---|---|---|---|---|---|---|---|---|---|
| Evelyn A. Brakema | 2019 | Kyrgyzstan | Cross-sectional | 2,050/750 | 26.0 ± 5.17 | Fixed ratio | 392 | 187 (47.7) | 47.2 ± 15.3 | 3 (0.8) | 106 (27.0) | 54 (13.8) | 7 | ①②③④⑤ |
| André's Caballero | 2008 | Colombian (2003.02–2004.05) | Cross-sectional | 1,538–2,640/18–995 | NA | Fixed ratio | 5,539 | 1,838 (33.2) | ≥40 | NA | 1,014 (18.3) | NA | 8 | ②③⑤⑥⑦ |
| Ana Maria B. Menezes | 2005 | Americas/Mexico | Cross-sectional | 2,240/35–950 | NA | Fixed ratio | 5,315 | 999 (18.8) | ≥40 | NA | 907 (17.1) | 579 (10.9) | 10 | ②③④⑤⑥⑧⑨ |
| Rafael Laniado-Laborin | 2012 | Mexico (2008.03–2008.10) | Cross-sectional | >2,000/0–10 | 28.3 ± 4.82 | Fixed ratio | 2,293 | NA | 55.3 ± 21.7 | NA | NA | NA | 6 | ②③⑤⑥ |
| Andreas Horner | 2017 | Austria (2003–2012) | Cross-sectional | >1,500/≤1,500 | NA | LLN | 30,874 | 13,644 (44.2) | 56.1 ± 11.3 | 957 (3.1) | 7,065 (22.9) | 6,725 (21.8) | 8 | ②③④⑤⑥⑧ |
| Catherine H. Miele | 2018 | Peru (2010.09–?) | Cross-sectional | 3,825/0–500 | 27.7 ± 4.6 | Fixed ratio | 3,048 | 1,499 (49.2) | 55.4 ± 12.5 | 89 (3.0) | 101 (3.3) | NA | 8 | NA |
| Devan Jaganath | 2015 | Peru (2010.09–?) | Cross-sectional | 3,825/0–500 | NA | Fixed ratio | 2,957 | 1,457 (49.3) | 45.2 ± 14.2 | 86 (2.9) | 97 (3.3) | NA | 9 | NA |
| Denis Vinnikov | 2011 | Kyrgyzstan (2005–2009) | Cross-sectional | 3,800–4500 | NA | Fixed ratio | 842 | 737 (87.5) | 38.9 ± 8.6 | NA | 449 (53.3) | NA | 7 | NA |
| S. Zaeh | 2016 | Peru (2010.09–?) | Cross-sectional | 3,825/0–500 | 27.8 ± 4.6 | Fixed ratio | 2,953 | 1,447 (49.0) | 55.3 ± 12.4 | NA | NA | 635 (21.5) | 8 | NA |
| Diego Urrunaga-Pastor | 2018 | Peru (2013–2016) | Cross-sectional | 2,158–3,847 | NA | Patient-reported | 365 | 123 (33.7) | ≥60 | NA | NA | 2 (0.6) | 5 | NA |

**Notes:**
BMI, Body Mass Index; NA, Not Applicable; TB, Tuberculosis; AHRQ, Agency for Healthcare Research and Quality; LLN, Lower Limit of Normal. Adjusted OR: ①, household air pollution; ②, age; ③, gender; ④, educational level; ⑤, pack years; ⑥, exposure to gas or dust; ⑦, history of tuberculosis; ⑧, body mass index; ⑨, ethnic origin.
*Diagnostic criteria*: Fixed ratio: post-bronchodilator forced expiratory volume in 1s (FEV1)/forced vital capacity (FVC) ratio was less than 0.70.
LLN: FEV1/FVC less than 5% of age-dependent lower limit of normal; Patient-reported COPD: based on the questions: "Did your doctor ever tell you that you had chronic bronchitis?" "Did a doctor ever tell you that you had emphysema?" "Have you ever been told by a doctor that you had chronic obstructive respiratory disease?"

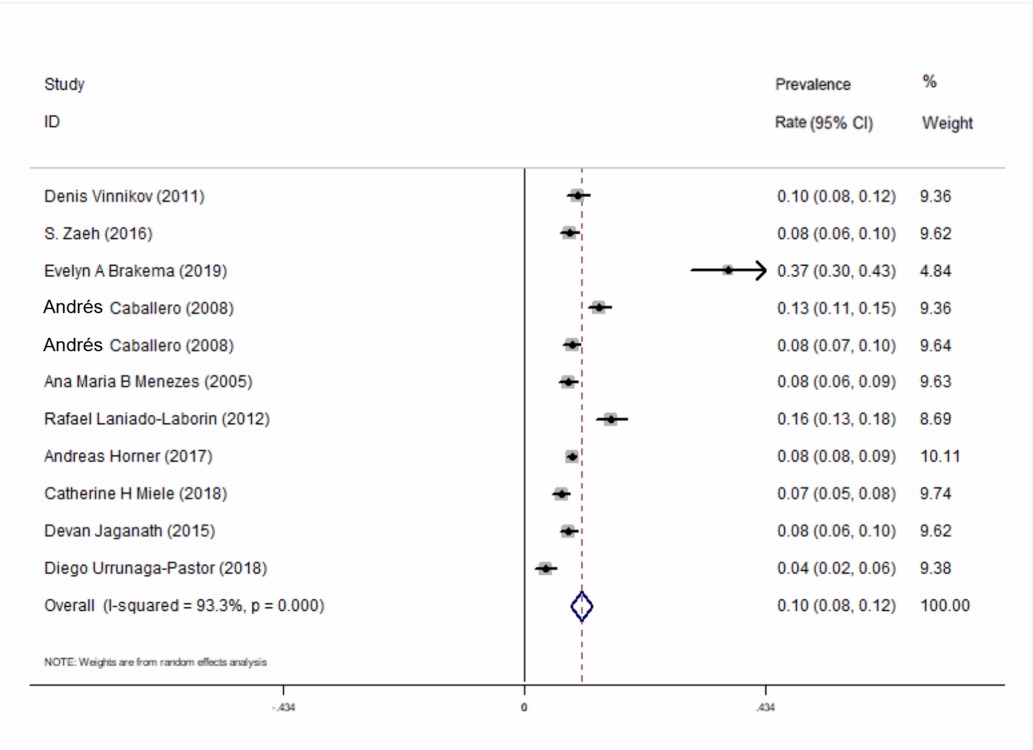

**Figure 2  Forest plot for the prevalence of COPD at high-altitude.**

(95% CI [0.08–0.12], $P < 0.001$) (Fig. 2). In a subgroup analysis based on different regions, the results showed that the prevalence in Asia was higher than that in the other 2 continents (Fig. 3). There was no publication bias among the studies (Begg's test $z = 1.34$, $P > 0.180$; Egger's test $P > 0.143$). A plot for publication bias is shown in Fig. S1. The result of the funnel plot is shown in Fig. S2.

Seven studies compared the relationship between the prevalence of COPD at high-altitudes and lowlands and calculated the adjusted OR for the altitude in COPD and that for non-COPD patients. The results showed that altitude is not an independent risk factor for the prevalence of COPD ($OR_{adj} = 1.28$, 95% CI [0.93–1.76], $P = 0.129$) (Fig. 4). Similar results were also found in the subgroup for different regions (Fig. S3).

# DISCUSSION

This meta-analysis summarized the global prevalence of COPD at high altitudes and analyzed whether altitude is one of the risk factors for the onset of high-altitude COPD. The main findings of this study were as follows: the prevalence of COPD at high altitude is 10.0%, which is higher than the global average (3.2 in males and 2.0 in females) (*Brakema et al., 2019*; *Miranda et al., 2012*). From the subgroup analysis, the prevalence of COPD at high altitude was found to be higher in Asia than that in the Americas and Europe. However, altitude was not found to be an independent risk factor for the occurrence of COPD.

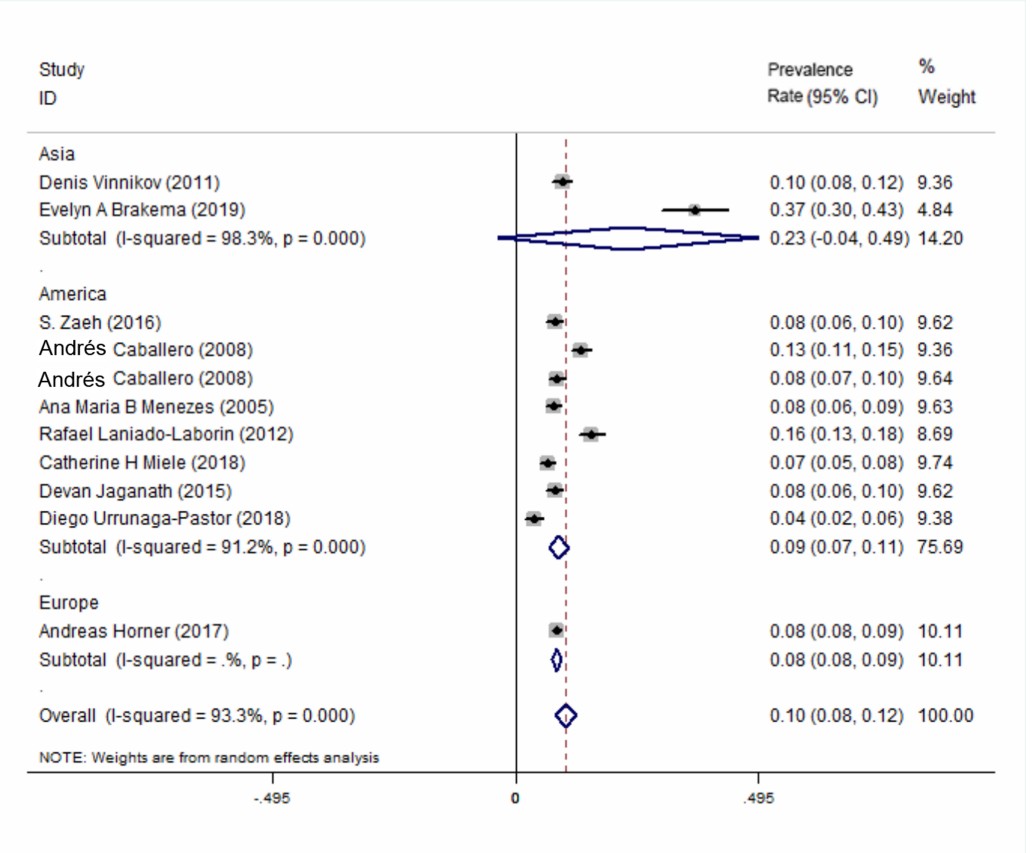

**Figure 3 Forest plot for the prevalence of COPD at high-altitude by different regions.**

High prevalence of COPD was found at high-altitudes, which was inconsistent with *Horner et al. (2017)*. Altitude was not found to be an independent risk factor for such phenomenon, which demonstrated that there might be other risk factors for COPD at high-altitudes. We tried to find out other possible risk factors from the original studies, but unfortunately, the information in the original studies was limited. A study reported that HAP might be a reason for this prevalence (*Gordon et al., 2014*). The study also found that people living in high-altitudes were more likely to be exposed to HAP than those living in the lowlanders. The possible mechanism for HAP leading to COPD was the change in the innate immune response, increase in inflammation in the lungs and oxidative stress state (*Olloquequi & Silva, 2016*). Moreover, a study by *Miele et al. (2018)* found that it might be due to the high-speed decline in lung function. The multivariable regression analysis in their study showed that people living at high altitudes could suffer an additional decline of lung function. Other possible reasons might be alveolar oxygen partial pressure ($PAO_2$), arterial oxygen partial pressure ($PaO_2$), temperature, and humidity. There is "physiological pulmonary hypertension" in healthy people living at high altitudes. Furthermore, low temperature and humidity at high altitudes may increase dryness in the airways, and reduce respiratory quality. Although there were many

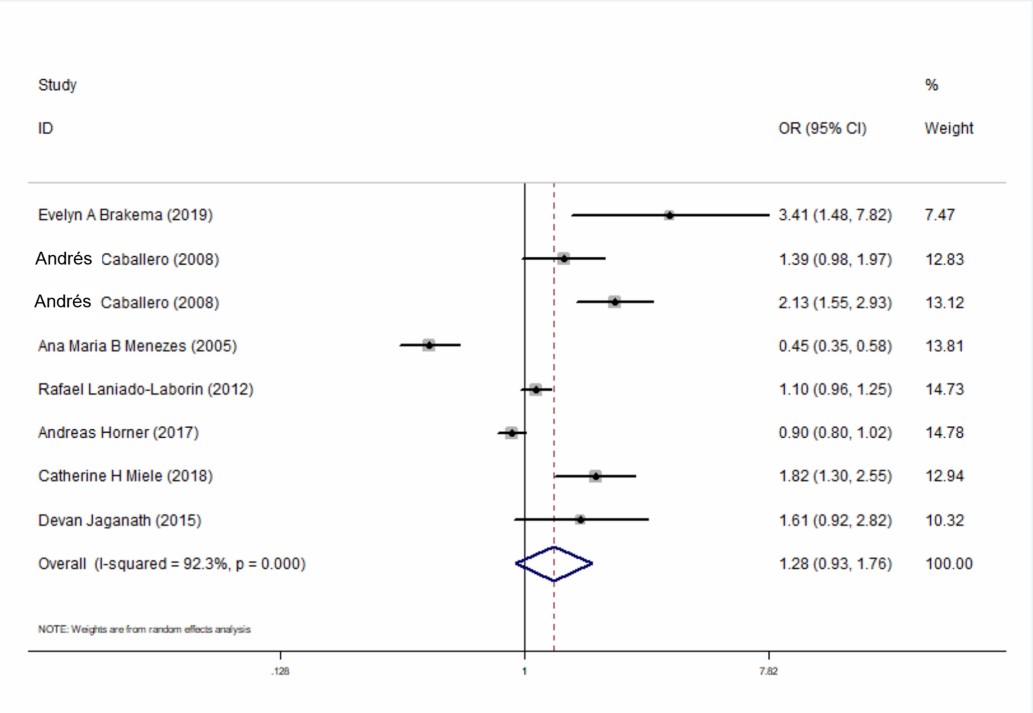

**Figure 4 Forest plot for assessing whether altitude is a risk factor for developing COPD.**

conjectures, the exact risk factors are still unknown. Therefore, it is essential for future studies to find out the exact factors that cause a high prevalence of COPD at high-altitudes.

A subgroup analysis according to different regions showed that the prevalence of COPD at high-altitudes in Asia was higher than that in the Americas and Europe. However, only a small number of studies included were from Asia and Europe, and this result had small representation.

The differences between the regions might be due to the different genetic makeup of different ethnic groups. The deficiency of alpha-1 antitrypsin (AAT) was the best known for developing COPD. Several studies investigated the prevalence of alpha1-antitrypsin deficiency (AATD), such as that by *De Serres (2002)*. They found that this disease was mainly found in Europe and was a common genetic disease in Caucasians and that there were still differences among countries. Some scholars have investigated and statistically analyzed the frequency of mutations among residents of 21 European countries (*Blanco et al., 2006*). They discovered that the Z-type mutations were mainly in the Nordic coastal areas and that the mutations in the S allele that caused moderate levels of plasma AAT were more common in southern Europe. The frequency of the Z mutant gene in America was lower than that in Europe, while the S mutation was higher in subjects from Northern Europe (*Blanco et al., 2006*). Recently, matrix metalloproteinase 12 (MMP-12) (*Hunninghake et al., 2009*) and glutathione S-transferase (GST) (*Ding et al., 2019*) were also found to be possible risk factors. A meta-analysis (*Ding et al., 2019*) from a Chinese group found that GSTM1 and GSTT1 deletion genotypes significantly increased

the prevalence of COPD risk. A racial subgroup analysis showed that the polymorphism of GSTM1 deletion gene was closely related to the susceptibility of COPD for all races and that the polymorphism of the GSTT1 deletion gene was only associated with COPD in Asia.

A subgroup analysis for detecting the source of heterogeneity showed that region was not a main source of heterogeneity. Heterogeneity for the prevalence of COPD was found to be due to several reasons. First, different diagnostic criteria were used in different studies. There were no gold diagnostic criteria for COPD. Spirometry was widely used in clinical trials because it is noninvasive and easy to perform. However, fixed-ratio criteria was highly under-diagnosed in patients under 45 years of age and over-diagnosed in the elderly (*Halbert et al., 2006*; *Vogelmeier et al., 2017*). This diagnostic criteria were largely influenced by the degree of cooperation of the subjects. The educational level of patients may have affected the coordination and understanding of spirometry. In our meta-analysis, 8 out of the 10 studies used fixed-ratio criteria to diagnose COPD, but the percentage of higher education varied from 0.6% to 21.8%. This might present a potential diagnostic heterogeneity. Second, different ages were included in the original studies. A study from Latin America by *Lamprecht et al. (2011)* examined the prevalence of COPD in subjects over 40 years of age (*Lamprecht et al., 2011*) and showed that the prevalence of COPD increased steadily with age, especially in those over 60 years of age. *Urrunaga-Pastor et al. (2018)* only included participants over 60 years in their study, which might have caused a higher prevalence of COPD than other studies. Therefore, age heterogeneity might have been a source of the overall heterogeneity. Third, a different portion of males was included in different studies. Although a study reported that the prevalence of COPD was equal in males and females (*Wood et al., 2003*), a meta-analysis from *Halbert et al. (2006)* demonstrated that there was a higher prevalence of COPD in males than females. However, the proportion of males in the included studies varied from 18.8% to 87.5%, this might have been a partial cause of the heterogeneity.

This meta-analysis had several limitations. First, all of our included participants were over the age of 40, while *Van Gemert et al. (2015)* said that people under the age of 30 who had been exposed to risk factors for a long time might be diagnosed as COPD. Therefore future studies on COPD should include younger participants. Second, no study described the difference in mortality of COPD between high-altitudes and lowlands. Third, we tried to find out whether the prevalence of COPD increased with an increase in altitude (*Ezzati et al., 2012*). However, unfortunately, we couldn't get exact data from the original studies, therefore, further studies are needed to support our conjecture. Fourth, most of the regions included in our study were from American countries, and only a few were from other continents, this might have led to the small representation of our results on a global scale. Finally, only one study reported that they have considered the ethnic origin as a mixed factor, and we could not rule out the impact of race on our results. As ethnic composition changes with altitude, often native ethnicity (American Indians) increases with altitude, that may change susceptibility.

## CONCLUSIONS

The results of this meta-analysis suggest that the prevalence of COPD at high-altitudes is higher than that from average data. However, altitude was not found to be an independent risk factor for developing COPD. It is essential for future studies to detect risk factors for the prevalence of COPD at high altitudes. Studies involving younger participants and larger geographical regions are important to make these studies more representative and reliable.

## ABBREVIATIONS

| | |
|---|---|
| AAT | alpha-1 antitrypsin |
| AATD | alpha1-antitrypsin deficiency |
| AHRQ | Agency for Healthcare Research and Quality |
| AMSTAR 2 | assessing the methodological quality of systematic reviews |
| ARDS | acute respiratory distress syndrome |
| BMI | body mass index |
| CI | confidence interval |
| COPD | chronic obstructive pulmonary disease |
| GST | glutathione S-transferase |
| HAP | house air pollution |
| MMP-12 | matrix metalloproteinase 12 |
| NOS | Newcastle-Ottawa Scale |
| OR | Odds Ratio |
| $PaO_2$ | arterial oxygen partial pressure |
| $PAO_2$ | alveolar oxygen partial pressure |
| TB | tuberculosis |

### Funding

The authors received no funding for this work.

### Competing Interests

The authors declare that they have no competing interests.

### Author Contributions

- Huaiyu Xiong conceived and designed the experiments, performed the experiments, prepared figures and/or tables, authored or reviewed drafts of the paper, and approved the final draft.
- Qiangru Huang conceived and designed the experiments, prepared figures and/or tables, and approved the final draft.
- Chengying He analyzed the data, prepared figures and/or tables, and approved the final draft.

- Tiankui Shuai performed the experiments, analyzed the data, prepared figures and/or tables, and approved the final draft.
- Peijing Yan conceived and designed the experiments, analyzed the data, prepared figures and/or tables, and approved the final draft.
- Lei Zhu performed the experiments, prepared figures and/or tables, and approved the final draft.
- Kehu Yang analyzed the data, authored or reviewed drafts of the paper, and approved the final draft.
- Jian Liu analyzed the data, authored or reviewed drafts of the paper, and approved the final draft.

## Data Availability

The raw data is available in the Supplemental File.

## Supplemental Information

Supplemental information for this article can be found online at http://dx.doi.org/10.7717/peerj.8586#supplemental-information.

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
