# Peer review of "Prevalence of chronic obstructive pulmonary disease at high altitude: a systematic review and meta-analysis"

_PeerJ, doi:10.7717/peerj.8586_

## Round 0.1 · original submission · Major Revisions

Thank you for a job well done. The comments are well crafted and provide a road map to revise and resubmit. If you address the comments directly in a revision, this should be an acceptable paper. Please note, one of the reviewers points out you limited your search to four databases. Please expand this and determine if there were papers you did not identify in your first search. Additionally, please note, you need to define high altitude with a specific definition. When you do, please note if this would have any impact on your search criteria.

There was also another review published recently, Horner A, Soriano JB, Puhan MA, et al. Altitude and COPD prevalence: analysis of the PREPOCOL-PLATINO-BOLD-EPI-SCAN study. Respiratory research 2017; 18. Please ensure that your work complements that paper or expands upon it.

Overall, a nice job and the revision should be straight forward. Good luck.

Reviewer 1 ·

Basic reporting

Basic reporting adequate, including components of the manuscript, professional structure, short manuscript.

Experimental design

Design is adequate, question clearly defined, impact of altitude in COPD prevalence is relevant from the scientific point of view and also from a practical point of view as there is a substantial proportion of individuals living at high altitude especially in some countries.
Limitation of the analysis derives from a limited number of original studies analyzing the question, although adding to a large number of individuals.

Methods are well described although not enough detail on obtaining adjusted OR in each study, and what variables were adjusted, what is important for the summary adjusted OR

Validity of the findings

The summary analysis is correct (except previous observation on adjusted OR´s with unclear variables adjusted for)

Additional comments

Interesting review on the impact of altitude on prevalence of COPD, analyzing crude and adjusted OR. Study well described and including all important items.
Crude OR suggests altitude is a risk factor, but adjusted OR did not, as there was confusing factors better explaining it. Not clear what variables were adjusted for in the individual studies (should be clarified)
A recent review analyzing almost the same original manuscripts. Horner A, Soriano JB, Puhan MA, et al. Altitude and COPD prevalence: analysis of the PREPOCOL-PLATINO-BOLD-EPI-SCAN study. Respiratory research 2017; 18, worked on individual data.

In several parts of the manuscript the authors talk about incidence but all studies analyzed are cross-sectional and the manuscripts describe prevalences.

Several limitations, as ethnic composition changes with altitude, often with native (amerindian in America) ethnicity increases with altitude, also socioeconomic status, use of biomass as a fuel, and others, that may change susceptibility.

Reviewer 2 ·

Basic reporting

The study “Prevalence of chronic obstructive pulmonary disease at high altitude: a systematic review and meta-analysis ” is a well-written article that identifies an important gap. However, to make this paper publishable, the author needs to respond to the following points.

Abstract.
Include the objective of the study.
If the objective was to analyze previous studies to conduct a meta-analysis and summarize the prevalence of COPD at high-altitudes and explore whether altitude is an independent risk factor affecting the prevalence of COPD; Why is the conclusion in terms of incidence?

Experimental design

Methods. The study "Prevalence of chronic obstructive pulmonary disease at high altitude: a systematic review and meta-analysis" is original and there are new data. Methods were described in detail, but there are some points that must be addressed:

1. The bibliographic search in four databases (PubMed/Medline, Cochrane Library, WoS, and Embase) is very limited. They should include other databases (SCOPUS, OVID, ScieLO, and LiLacs).
2. What was the definition of high altitude?
3. Was high altitude defined as >1500 m above sea level?

Validity of the findings

Results:
1. The analyses are robust, and statistically sound.
2. If the authors included cross-sectional (prevalence) studies. Why were determined the results and the conclusion in terms of incidence?
3. I suggest include Mexico in Table 1 study of Ana Maria Menezes/Country “Americas”
4. For a better understanding of the readers, in the Forest plot (Figures 2, 3 and 4) include the prevalence rates (values) of COPD.
5. For in study of Andre´s Caballero (Colombia), how they have synthesized the prevalence rates for COPD in Medellin and Bogotá, (height 1538 and 2640 m, respectively).

Discussion:
1. Revise. Line 175. “...incidence of COPD at high altitude…” in results is as prevalence.
2. The conclusions should be appropriately stated in terms of prevalence (not incidence)

Additional comments

The manuscript is clearly written. If there are comments as I have noted above, which should be improved upon before Acceptance.

---

## Round 0.2 · accepted · Accept

Thank you for a full faith attempt to meet the comments of the reviewers in your revision. Thank you for adding specificity, clarity, and detail to your revision. Nice work. Congratulations.